

# Theoretical insights into the molecular mechanism of I117V mutation in neuraminidase mediated reduction of oseltamivir drug susceptibility in A/H5N1 influenza virus

Mohini Yadav[1], Manabu Igarashi[2,3] and Norifumi Yamamoto[1]

[1] Department of Applied Chemistry, Faculty of Engineering, Chiba Institute of Technology, Narashino, Japan
[2] Division of Global Epidemiology, International Institute for Zoonosis Control, Hokkaido University, Sapporo, Japan
[3] International Collaboration Unit, International Institute for Zoonosis Control, Hokkaido University, Sapporo, Japan

Corresponding author
Norifumi Yamamoto,
norifumi.yamamoto@it-chiba.ac.jp

## ABSTRACT

The substitution of Ile to Val at residue 117 (I117V) of neuraminidase (NA) reduces the susceptibility of the A/H5N1 influenza virus to oseltamivir (OTV). However, the molecular mechanism by which the I117V mutation affects the intermolecular interactions between NA and OTV has not been fully elucidated. In this study, we performed molecular dynamics (MD) simulations to analyze the characteristic conformational changes that contribute to the reduced binding affinity of NA to OTV after the I117V mutation. The results of MD simulations revealed that after the I117V mutation in NA, the changes in the secondary structure around the mutation site had a noticeable effect on the residue interactions in the OTV-binding site. In the case of the WT NA-OTV complex, the positively charged side chain of R118, located in the β-sheet region, frequently interacted with the negatively charged side chain of E119, which is an amino acid residue in the OTV-binding site. This can reduce the electrostatic repulsion of E119 toward D151, which is also a negatively charged residue in the OTV-binding site, so that both E119 and D151 simultaneously form hydrogen bonds with OTV more frequently, which greatly contributes to the binding affinity of NA to OTV. After the I117V mutation in NA, the side chain of R118 interacted with the side chain of E119 less frequently, likely because of the decreased tendency of R118 to form a β-sheet structure. As a result, the electrostatic repulsion of E119 toward D151 is greater than that of the WT case, making it difficult for both E119 and D151 to simultaneously form hydrogen bonds with OTV, which in turn reduces the binding affinity of NA to OTV. Hence, after the I117V mutation in NA, influenza viruses are less susceptible to OTV because of conformational changes in residues of R118, E119, and D151 around the mutation site and in the binding site.

## INTRODUCTION

Influenza A viruses infect a variety of avian and mammalian species, including humans (*Webster et al., 1992*). Influenza A viruses are divided into subtypes based on antigenic differences of two virus surface glycoproteins, hemagglutinin (HA) and neuraminidase (NA) (*Gamblin & Skehel, 2010*). A total of 16 HA (H1–H16) and nine NA (N1–N9) subtypes have been isolated from wild waterfowl so far (*Fouchier et al., 2005*). HA mediates virus entry into the host cell by binding to a terminal sialic acid on the host cell surface. NA is responsible for removing sialic acid to facilitate the release of progeny viruses from infected cells. Several NA inhibitors, such as oseltamivir (OTV), zanamivir, laninamivir, and peramivir, are currently available for the treatment of influenza virus infection (*McKimm-Breschkin, 2012*). Among them, OTV is the most widely used anti-influenza drug (*Kim et al., 1997*).

OTV-resistant H1N1 and H5N1 viruses have been isolated from humans as well as avian or swine species (*Monto et al., 2006*; *Rameix-Welti et al., 2006*; *McKimm-Breschkin et al., 2007*). This suggests that viruses could acquire reduced sensitivity to OTV not only by drug-selective pressure but also by natural genetic variation. In the mid-2000s, several H5N1 viruses with an Ile-to-Val substitution at position 117 of NA (I117V) were isolated from some avian species (*Hurt et al., 2007*; *McKimm-Breschkin et al., 2007*; *Govorkova et al., 2009*; *Chen et al., 2010*; *Ilyushina et al., 2010*; *McKimm-Breschkin et al., 2013*; *Takano et al., 2013*; *Marinova-Petkova et al., 2014*; *Creanga et al., 2017*; *Kode et al., 2019*). *In vitro* and *in vivo* experiments have shown that the I117V mutant NA conferred a reduction in susceptibility to OTV as compared to the wild-type (WT) (*Hurt et al., 2007*; *McKimm-Breschkin et al., 2007*; *Chen et al., 2010*; *Ilyushina et al., 2010*; *McKimm-Breschkin et al., 2013*; *Takano et al., 2013*; *Creanga et al., 2017*; *Kode et al., 2019*). Interestingly, residue 117 is not contained in the drug-binding site of NA, which consists of eight functional residues (R118, D151, R152, R224, E276, R292, R371, and Y406; N2 numbering) and 11 framework residues (E119, R156, W178, S179, D198, I222, E227, H274, E277, N294, and E425; N2 numbering) (*Colman, Varghese & Laver, 1983*; *Colman, Hoyne & Lawrence, 1993*). The molecular mechanism underlying how the mutation of residue 117, which is not part of the drug-binding site of NA, indirectly affects the molecular interaction between NA and OTV has not been fully elucidated.

Several computational studies using molecular dynamics (MD) simulations have reported on the molecular mechanism of reduced susceptibility to OTV in the I117V mutant (*Takano et al., 2013*; *Mhlongo & Soliman, 2015*). *Takano et al. (2013)* evaluated the effects of the I117V mutation in NA on OTV susceptibility *in vitro*, *in vivo*, and *in silico*. Their experimental results showed that the I117V mutation caused a slight reduction in the susceptibility of NA to OTV *in vitro* and dramatically *in vivo*. They also analyzed a single 2.5-ns trajectory obtained from MD simulations to further investigate the molecular mechanism by which the I117V mutation reduces the susceptibility of NA to OTV. Their computational results showed that the I117V mutation decreased the binding affinity for OTV because of the loss of hydrogen bonds between the carboxyl group of OTV and the side chain of R118 of NA. *Mhlongo & Soliman (2015)*

analyzed four distinctive 25-ns trajectories obtained from MD simulations to investigate the molecular mechanism of the reduced susceptibility of the I117V mutant NA to OTV. Their computational results showed that the I117V mutation distorts the orientation of OTV in the drug-binding site of NA because of the loss of hydrogen bonds between the amino group of OTV and the side chain of E119 of NA, resulting in reduced binding affinity of NA to OTV. In these previous computational studies, the production trajectories of the MD simulations were too short to reach reliable statistical results. In addition, they focused on changes in the direct interactions between OTV and amino acid residues in the drug-binding site of NA. However, it was not clear how the I117V mutation of NA at a point outside its drug-binding site could cause changes in the intermolecular interaction with OTV.

In this study, we performed four distinctive 100-ns MD simulations for the WT and I117V mutant NA-OTV complexes in the A/H5N1 influenza virus. Based on the multiple production trajectories obtained from MD simulations, we analyzed the characteristic conformational changes around the I117V mutation site of NA, which greatly affected the intermolecular interactions with OTV. The results showed that after the I117V mutation in NA, the binding affinity between NA and OTV was reduced due to the conformational change of R118 adjacent to the mutation site, which affected the interactions of E119 and D151 with OTV. Thus, the present study successfully clarified the molecular mechanism by which the I117V mutation reduces the susceptibility of NA to OTV.

## METHODS

### Initial structures

The coordinate of WT avian influenza virus A/H5N1 NA in complex with OTV was obtained from the Protein Data Bank (PDB code: 2HU4) (*Russell et al., 2006*). The complex structure of the I117V mutant NA and OTV was generated by replacing isoleucine (Ile) 117 in the WT complex with valine (Val). H5N1 NA contains one calcium ion, which is necessary for structural stability (*Smith et al., 2006*), but no calcium ions were found in the crystal structure registered as 2HU4. The coordinate of the calcium ion in NA was obtained from the structure of A/H5N1 NA registered as 3CL0 (*Collins et al., 2008*). The protonation state of histidine (His) in NA at pH 7 was determined using the PDB2PQR server (*Dolinsky et al., 2004*). The other ionized residues, such as arginine (Arg), lysine (Lys), aspartic acid (Asp), and glutamic acid (Glu), were treated as charged entities. The missing hydrogen atoms in NA and OTV were added using the LEaP program in the Amber 20 package (*Case et al., 2020*). For each disulfide bond in NA, a covalent bond was created between the proximate cysteine residues using the LEaP program. The FF14SB variant of the AMBER force field was used to describe NA (*Maier et al., 2015*). The parameters of the generalized AMBER force field (GAFF) were applied to OTV (*Wang et al., 2004*). The partial atomic charges in OTV were determined on the basis of *ab initio* quantum chemistry calculations at the HF/6-31G(d) level with the Gaussian 16 program package (*Frisch et al., 2016*), following the restrained electrostatic potential fitting procedure (*Bayly et al., 1993*). The complexes of NA and

OTV were dissolved in a truncated octahedral box filled with water molecules, where the box size was set so that there was a distance of at least 10 Å between the complexes and the boundary of the box. The TIP3P model was used to represent water molecules (*Jorgensen et al., 1983*). The total charge of the systems was neutralized by the addition of sodium counter ions. Periodic boundary conditions were adopted.

## MD simulations

MD simulations were performed using the PMEMD module in the Amber 20 package (*Case et al., 2020*). The geometry of each system was optimized (energy minimized) using the steepest descent algorithm for 500 steps, followed by the conjugate gradient algorithm for 4,500 steps. After geometry optimization, each system was heated until the temperature ($T$) reached 300 K over a period of 200 ps in the *NVT* ensemble, while applying a harmonic restraint of 2 kcal mol$^{-1}$ Å$^{-2}$ on the complexes of NA and OTV, except for the hydrogen atoms. The temperature was regulated using the weak-coupling algorithm (*Berendsen, Postma & Funsteren, 1984*). After heating, 10 ns of MD simulations were performed to equilibrate the system in the *NpT* ensemble at $T$ = 300 K and a pressure ($p$) of 1.0 atm. The pressure was maintained using a Berendsen barostat. After equilibration, additional 10-ns MD simulations were performed in the *NpT* ensemble at $T$ = 300 K and $p$ = 1.0 atm. During MD simulations, all covalent bond lengths were constrained using the SHAKE algorithm (*Ryckaert, Ciccotti & Berendsen, 1977*). The time step of MD simulations was set to 2 fs. A cutoff for the non-bonded intermolecular interactions was set to 8 Å. Long-range electrostatic interactions were treated using the particle-mesh Ewald method (*Darden, York & Pedersen, 1993*). Finally, four copied MD simulations were performed for 100 ns starting with different coordinates and velocities in the *NpT* ensemble at $T$ = 300 K and $p$ = 1.0 atm, where the initial coordinates were randomly selected from the additional 10-ns trajectories after equilibration and the initial velocities were randomly reassigned. The production phase to be analyzed was the last 80 ns of MD simulations, which was determined based on the root mean square displacement (RMSD) of the backbone atoms in the proteins with respect to the initial structure along the simulation time. The time series of RMSD and radius of gyration for the backbone atoms in the WT and I117V mutant NA are shown in Figs. S1 and S2. The changes in RMSD were almost constant after 20 ns, indicating that the MD simulations properly converged in the region of 20–100 ns.

## Binding free energy calculations

Binding free energies were determined for 400 frames extracted from the four distinctive production phases of the MD simulations, based on the Molecular Mechanics Poisson Boltzmann Surface Area (MM-PBSA) continuum solvation method (*Kollman et al., 2000*). The MM-PBSA calculations were performed using the MMPBSA.py program in the Amber 20 package (*Miller et al., 2012*; *Case et al., 2020*). The adaptive Poisson Boltzmann (PB) solver was used to estimate the electrostatic contribution to the solvation free energy (*Baker et al., 2001*). The linear PB equation was solved using a maximum of 1,000 iterations. The surface area for the nonpolar solvation energy term was determined using

the Linear Combination of Pairwise Overlap (LCPO) algorithm (*Weiser, Shenkin & Still, 1999*). In calculations using continuum methods, the dielectric properties of the protein interior and solvent are represented in terms of the dielectric constants. In this study, the dielectric constant of the protein interior was set to four, as a relatively large dielectric constant is desirable for NA, considering that its binding site contains many charged residues (*Hou et al., 2011*). The dielectric constant of the solvent phase was set to 80. The ionic strength was set at 150 mM. The ratio between the longest dimension of the rectangular finite-difference grid and that of the solute was set to four.

Entropies due to the vibrational degrees of freedom were calculated for 100 configurations by normal mode analysis using the NAB program in the Amber 20 package (*Case et al., 2020*). The geometry of each configuration was optimized (energy minimized) with a generalized Born solvent model, using a maximum of 10,000 steps with a target root-mean-square gradient of $10^{-3}$ kcal mol$^{-1}$ Å$^{-1}$.

## RESULTS

### Binding structures and energies

Figure 1 shows the snapshot images obtained from the MD simulations for the WT and I117V mutant NA-OTV complexes, which show the OTV binding site and the region adjacent to residue 117. As shown in Fig. 1, OTV bound to the WT or I117V mutant NA by forming hydrogen bonds with two negatively charged residues, E119 and D151, and three positively charged residues, R152, R292, and R371. Residue R118 has a positively charged side chain similar to R292 and R371 in the binding site, but no hydrogen bond formation with OTV was observed. This is supported by the co-crystal structure of WT A/H5N1 NA with OTV (PDB code: 2HU4) (*Russell et al., 2006*) showing that R118 is not in a position to form hydrogen bonds with OTV.

Table 1 summarizes the computed binding free energies ($\Delta G$) of OTV for the WT and I117V mutant NA obtained from the MM-PBSA calculations, along with the enthalpy ($\Delta H$) and entropy ($T\Delta S$). The binding free energies of OTV were computed to be −14.60 and −11.88 kcal mol$^{-1}$ for the WT and I117V mutant NA, respectively. The 2.72 kcal mol$^{-1}$ increase in the binding free energy of OTV due to the I117V mutation could slightly reduce the susceptibility of this inhibitor to NA. This is supported by the fact that the I117V mutant NA has a 3- to approximately 50-fold decrease in the relative susceptibility to OTV compared with the WT NA in H5N1 viruses (*Hurt et al., 2007*; *McKimm-Breschkin et al., 2007*; *Chen et al., 2010*; *Ilyushina et al., 2010*; *McKimm-Breschkin et al., 2013*; *Takano et al., 2013*; *Creanga et al., 2017*; *Kode et al., 2019*). According to the WHO's antiviral working group criteria, influenza A viruses with <10-fold change in the half maximal inhibitory concentration (IC$_{50}$) value were characterized as exhibiting normal inhibition, while those with 10- to 100-fold and >100-fold changes exhibited reduced and highly reduced inhibition, respectively (*World Health Organization, 2012*). If experiments are done under the same conditions, the relative binding free energy of $\Delta\Delta G = \Delta G^{(1)} - \Delta G^{(2)}$ can be approximated using $\Delta\Delta G \cong RT$ ln (IC$_{50}^{(1)}$/IC$_{50}^{(2)}$), where $R$ is the ideal gas constant and $T$ is the temperature. The experimentally observed 3- to 50-fold change in the IC$_{50}$ value after I117V mutation

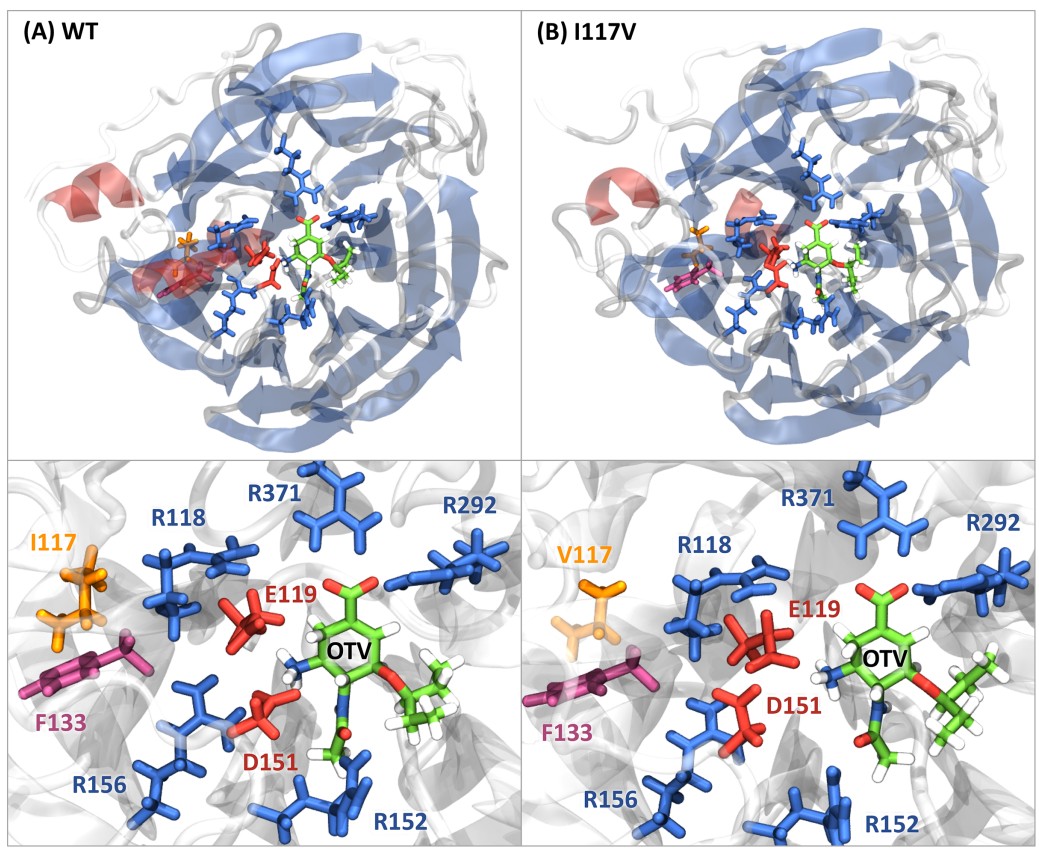

**Figure 1** **Snapshot images of the neuraminidase-oseltamivir complexes.** (A) Wild-type (WT) and (B) I117V mutant NA-OTV complexes. The upper panels show the overall structure of the complex in the cartoon representation, where the helix and sheet parts are colored in red and blue, respectively. The lower panels show the drug-binding site and the region adjacent to residue 117, where the positively charged residues (R118, R152, R156, R292, and R371) in the OTV binding site are represented in blue, and the negatively charged residues (E119 and D151) are represented in red.

**Table 1** **Calculated binding free energies (in kcal mol$^{-1}$) for oseltamivir to the wild-type (WT) and I117V mutant influenza neuraminidase obtained from the MM-PBSA calculations.**

|        | $\Delta H$        | $T\Delta S$       | $\Delta G$        | $\Delta\Delta G$ |
| ------ | ----------------- | ----------------- | ----------------- | ---------------- |
| WT     | −33.78 ± 0.08     | −19.18 ± 1.15     | −14.60 ± 1.23     |                  |
| I117V  | −31.06 ± 0.08     | −19.18 ± 1.20     | −11.88 ± 1.28     | 2.72             |

corresponds to a binding free energy difference of 0.7−2.3 kcal mol$^{-1}$. The current results are qualitatively consistent with the experimental studies, indicating that the MD simulations, which form the basis for subsequent analyses, are reliable.

In this study, we adopted the single-trajectory approach in the MM-PBSA calculation, because it assumes that no significant conformational changes occur upon ligand binding. The single-trajectory MM-PBSA approach has been widely used in previous studies, to determine binding free energy differences because of its good balance between computational cost and reliability (*Wang et al., 2019*). In some cases the single-trajectory

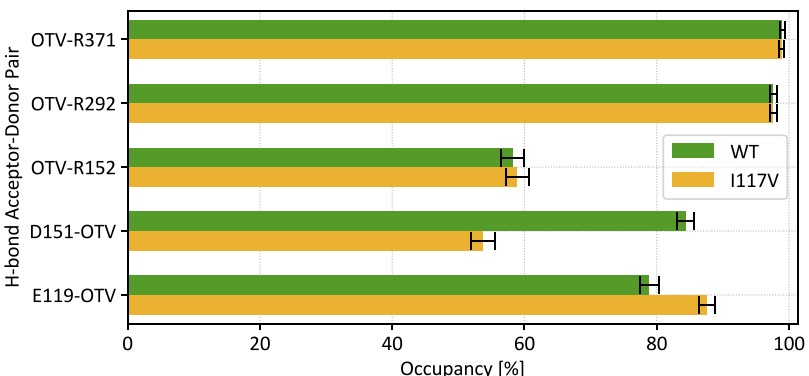

**Figure 2 Hydrogen bond occupancies of oseltamivir for the wild-type (WT) and I117V mutant neuraminidases.** In hydrogen bonding with OTV, the side chains of E119 and D151 acted as hydrogen acceptors, while the side chains of R152, R292, and R371 acted as hydrogen donors. Error bars represent 95% confidence intervals.           

approach used in this study is less reliable for determining binding free energies than the multiple-trajectory approach that accounts for conformational changes upon drug binding. However, we emphasize that the binding free energy difference of 2.72 kcal mol$^{-1}$ between the WT and I117V mutant NA determined in the present study is in good agreement with the experimentally determined value of 0.7–2.3 kcal mol$^{-1}$, which reveals that our MD simulations are sufficiently reliable for analyzing changes in various intra-protein interactions, such as hydrogen bonding, and secondary structures, which is the focus of this study.

As shown in Fig. 1, residue 117 did not interact directly with OTV in either WT or I117V mutant NA. Thus, the decrease in binding affinity between OTV and NA due to the I117V mutation could be the result of indirect effects due to changes in the interaction network of amino acid residues inside the protein. As shown in Table 1, the change in the entropic component ($T\Delta S$) upon I117V mutation is almost zero, which indicates that the difference in the binding free energies ($\Delta\Delta G$) is mostly enthalpy-driven rather than entropy-driven. Based on this observation, further analyses that focus on the factors responsible for changes in the direct interactions between OTV and NA are expected to be helpful. To elucidate the molecular mechanism by which the I117V mutation of NA at a point outside its drug-binding site could reduce the susceptibility to OTV, we performed the following detailed analysis based on the results of MD simulations.

## Hydrogen bond analysis

Figure 2 shows the hydrogen bond occupancies of OTV for the WT and I117V mutant NAs during the MD simulations. The standard errors in hydrogen bond occupancies are small (less than 1%); the 95% confidence intervals for hydrogen bond occupancies are shown as error bars in Fig. 2. Hydrogen bonds were assigned using PYTRAJ (*Nguyen et al., 2016*), a Python front-end package of the CPPTRAJ program (*Roe & Cheatham, 2013*). As shown in Fig. 2, OTV bound to NA by forming hydrogen bonds with five charged amino acid residues, E119, D151, R152, R292, and R371. In hydrogen bonding with OTV, the side chains of E119 and D151 acted as hydrogen acceptors, while the side chains of
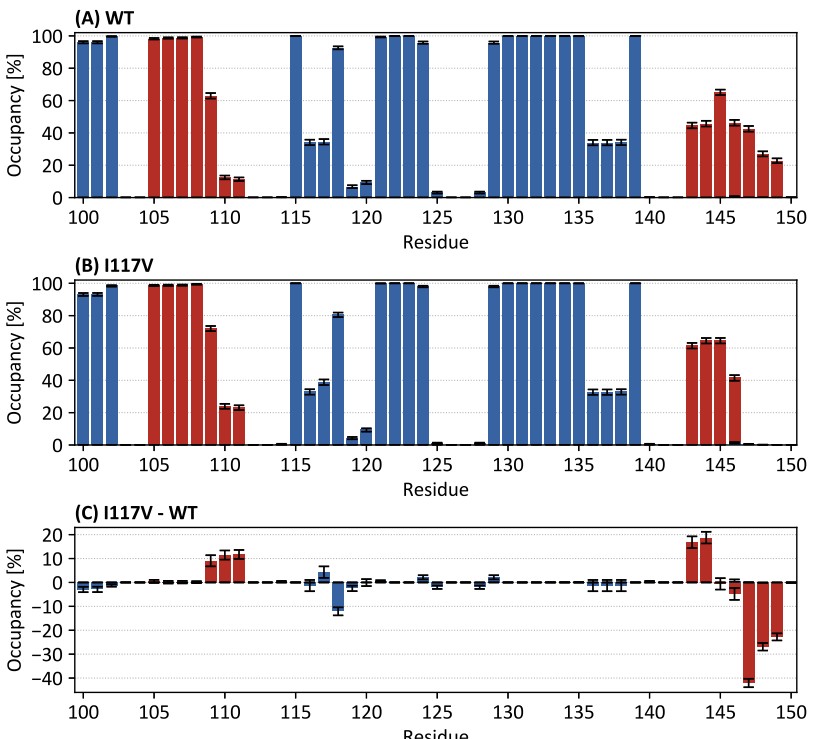

**Figure 3 Secondary structure occupancies in the region containing the 100th to 150th residues for the neuraminidase-oseltamivir complexes.** (A) The wild-type (WT) and (B) I117V mutant NA-OTV complexes. (C) Change in the secondary structure occupancies due to the I117V mutation. The secondary structures were classified into three simplified categories: helix, sheet, and coil. The helix and sheet components are represented by red and blue bars, respectively, while the rest correspond to the coil. Error bars represent 95% confidence intervals.

R152, R292, and R371 acted as hydrogen donors. The occupancies of the hydrogen bonds formed by R292 and R371 with OTV were almost 100% in both the WT and I117V mutant NA-OTV complexes, indicating that the interactions were extremely stable. The hydrogen bond formed between OTV and R152 was found to be relatively unstable, with an occupancy of approximately 60% in both WT and I117V mutant cases. Notable changes caused by the I117V mutation in NA were observed in the hydrogen bonds formed by E119 and D151 with OTV. Because of the I117V mutation in NA, the hydrogen bond occupancy of the E119-OTV pair increased by approximately 10%, whereas the hydrogen bond occupancy of the D151-OTV pair decreased by approximately 30%. D151 formed hydrogen bonds with the adjacent positively charged R156 amino acid residue when not bound to OTV. Thus, the instability of the hydrogen bond with D151 after the I117V mutation might be the major reason for the reduced binding affinity of NA to OTV.

## Secondary structure analysis

Figures 3A and 3B show the secondary structure occupancies in the region containing the 100th to 150th residues of NA for the WT and I117V mutant NA-OTV complexes, respectively. Figure 3C shows the changes in secondary structure occupancy after the I117V mutation. The secondary structure occupancies for all the residues in the NA are

shown in Fig. S3. The standard errors for secondary structure occupancies are small (less than 1%); the 95% confidence intervals for secondary structure occupancies are shown as error bars in Figs. 3A, 3B and 3C. The secondary structures were classified into three simplified categories (helix, sheet, and coil) using the PYTRAJ package (*Nguyen et al., 2016*) based on the DSSP program (*Kabsch & Sander, 1983*). The occupancies of secondary structures were calculated based on the assignment results for 3,200 three-dimensional structures extracted from four distinctive 80-ns trajectories in the production phase of MD simulations. In Fig. 3, the helix and sheet components are represented by red and blue bars, respectively, while the rest correspond to the coil.

As shown in Fig. 1, NA has an overall β-sheet-rich structure with partial helices. In the WT NA, as shown in Fig. 3A, the helix moieties were found in the region from the 105th to 111th residues and from the 143rd to 149th residues of NA. The 117th residue of interest in this study was located in the β-sheet region formed by residues between the 115th and 124th residues. After the I117V mutation, the secondary structure of the NA was significantly altered.

As shown in Fig. 3C, the occupancy of the helix component was significantly reduced in the region between residues the 147th and 149th residues with the I117V mutation. The secondary structure near the mutation site was also changed due to the I117V mutation, indicating that the β-sheet occupancy of R118 was reduced by approximately 12%. This may be due to a change in the orientation of R118 caused by the mutation of the bulkier Ile to the smaller Val at the 117th residue, which reduces the hydrogen-bonding interaction with residue L134 located in the adjacent antiparallel β-sheet moiety. Such conformational changes of R118 at a point inside the drug-binding site of NA would lead to a decrease in the binding affinity of the I117V mutant for OTV, due to the indirect effect of the Ile-to-Val mutation at residue 117.

## Residue-residue and residue-drug interactions

Figure 4 shows the correlations between the distances of the R118-E119 pair ($R_{R118-E119}$) and the D151-OTV pair ($R_{D151-OTV}$) in the WT and I117V mutant NA-OTV complexes as scatter plots and probability densities. The value of $R_{R118-E119}$ was determined by measuring the inter-atomic distance between the carbon atom in the guanidino group of R118 and the carbon atom in the carboxyl group of E119. The value of $R_{D151-OTV}$ was determined by measuring the inter-atomic distance between the carbon atom in the carboxyl group of D151 and the nitrogen atom in the amino group of OTV. Figure 5 shows the conformational fluctuations of the OTV binding site and adjacent I117V mutation site in the WT and I117V mutant NA-OTV complexes by superimposing 100 snapshot images obtained from the MD simulations.

In the case of the WT NA-OTV complex, as shown in Fig. 4, the distribution of $R_{D151-OTV}$ was generally localized in a monomodal manner around 3.6 Å, indicating that D151 tends to interact with OTV by forming hydrogen bonds. On the other hand, the distribution of $R_{R118-E119}$ was bimodal, with one strongly localized around 4.2 Å and the other weakly distributed around 6.0 Å, indicating that the side chains of R118 and E119 tended to interact closely, but were sometimes too far apart to interact. These characteristic
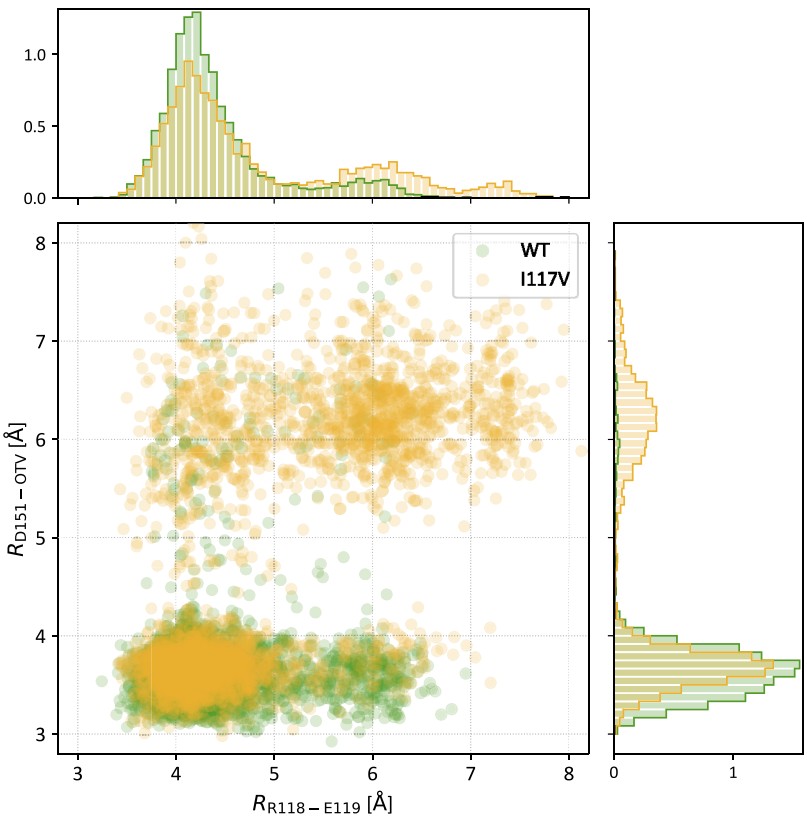

**Figure 4 Correlations between residue-residue and residue-drug interactions.** Scatter plots and probability densities for the distance of the R118-E119 pair ($R_{R118-E119}$) *versus* the distance of the D151-OTV pair ($R_{D151-OTV}$) in the wild-type (WT) and I117V mutant NA-OTV complexes.

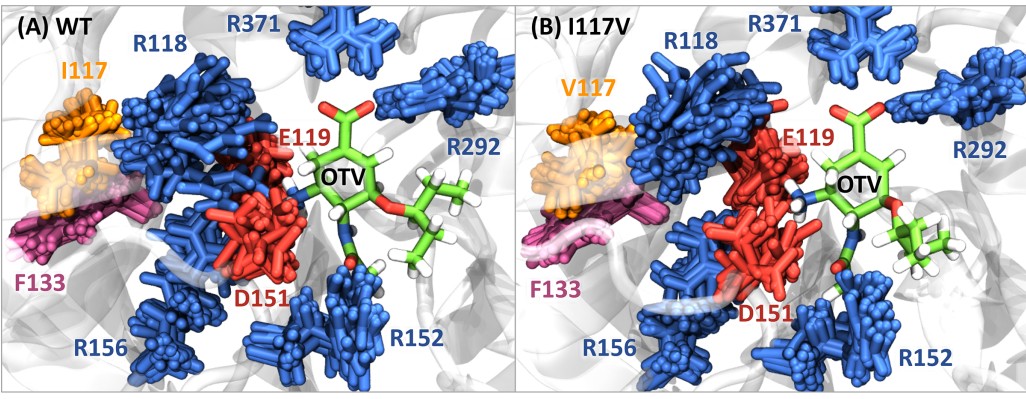

**Figure 5 Superimposed snapshot images for the neuraminidase-oseltamivir complexes.** (A) Wild-type (WT) and (B) I117V mutant NA-OTV complexes. OTV is represented in green, and residue 117 is indicated in orange. The positively charged residues (R118, R152, R156, R292, and R371) in the OTV binding site are represented in blue, and the negatively charged residues (E119 and D151) are represented in red.

conformational fluctuations of R118, E119, and D151 can also be seen in the snapshot images of the three-dimensional structure shown in Fig. 5A.

In the case of the I117V mutant NA-OTV complex, as shown in Fig. 4, the distribution of $R_{\text{D151-OTV}}$ was bimodal, such that in addition to the peak observed around 3.6 Å, a peak also appeared around 6.2 Å, unlike in the WT case. This indicates that, after the I117V mutation, the frequency of hydrogen bond formation between D151 and OTV was reduced, which is also supported by the results shown in Fig. 2. The relatively large peak around 6.2 Å also suggests that the 150 loop of NA was frequently opened after the I117V mutation, similar to what has been observed in the NA mutants of many drug-resistant strains (*Han, Liu & Mu, 2012*; *Kar & Knecht, 2012*; *Woods et al., 2012*; *Schaduangrat et al., 2016*; *Yadav, Igarashi & Yamamoto, 2021*). The distribution of $R_{\text{R118-E119}}$ was multimodal, with peaks near 4.2 Å and 6.0 Å as in the WT case, and an additional weak peak appearing near 7.4 Å. Here, compared to the WT case, the probability density of the main component at around 4.2 Å decreased, while that of the components at around 6.0 Å and 7.4 Å increased. This shows that after the I117V mutation, the side chains of R118 and E119 tended to separate frequently, thus not interacting with each other, compared to the WT case.

## DISCUSSION

### WT NA-OTV complex

In the WT NA-OTV complex, the residue-residue interaction between R118 and E119 may play a key role in enhancing the residue-drug interaction between D151 and OTV to increase the binding affinity of the WT NA to OTV. As shown in Figs. 1A and 5A, the negatively charged E119 interacts with the positively charged amino group of OTV, together with the negatively charged D151. Here, E119 and D151 tend to approach each other when interacting with OTV simultaneously, but the closer they are, the stronger the electrostatic repulsion between the negatively charged side chains. However, as shown in Fig. 2, the hydrogen bond occupancy of the E119-OTV and D151-OTV pairs was approximately 80% for both, indicating that E119 and D151 can form relatively stable hydrogen bonds with OTV. As shown in Fig. 3A, R118 is in the β-sheet region, indicating that its side chain can be rigidly oriented. Owing to the strong directivity of R118, derived from its secondary structure formation, its positively charged side chain can frequently interact in parallel with the negatively charged side chain of the adjacent E119. When R118 and E119 interact, the positive and negative charges of their side chains neutralize each other, thereby suppressing the electrostatic repulsion between E119 and D151. Thus, both E119 and D151 can simultaneously form hydrogen-bonding interactions with OTV, which contributes to the enhancement of the binding affinity of NA to OTV.

### I117V mutant NA-OTV complex

In the I117V mutant NA-OTV complex, the binding affinity of NA to OTV may be reduced by the weakening of the residue-drug interaction between D151 and OTV, accompanied by a decrease in the opportunity for residue-residue interaction between R118 and E119. As shown in Fig. 3C, the occupancy of R118 forming the β-sheet structure

decreased after the I117V mutation, indicating that the directionality of its side chain was weakened. The weakening of the directionality of its positively charged side chain reduces the opportunity for interaction with the negatively charged side chain of the adjacent E119. The reduced interactions between the side chains of R118 and E119 are shown in Fig. 4. As mentioned earlier, in the WT NA-OTV complex, the interaction between R118 and E119 can contribute to reducing the electrostatic repulsion between E119 and D151. However, in the I117V mutant NA-OTV complex, the electrostatic repulsion between E119 and D151 can be enhanced, since R118 has less opportunity for interaction with E119. This inhibits both E119 and D151 from simultaneously forming hydrogen-bonding interactions with the same positively charged amino group of OTV, resulting in a decrease in the binding affinity between NA and OTV. Thus, the change in the interactions of these residues after the I117V mutation slightly reduces the binding affinity of NA to OTV, resulting in a reduction in OTV drug susceptibility to influenza viruses.

As mentioned in the Introduction, several computational studies that use MD simulations have reported on the molecular mechanism associated with the lower susceptibility of the I117V mutant to OTV (Takano et al., 2013; Mhlongo & Soliman, 2015). Takano et al. (2013) analyzed a single 2.5-ns MD trajectory to show that the loss of hydrogen bonding between the R118 side chain in NA and the OTV carboxyl group after I117V mutation is responsible for the reduced susceptibility of NA to OTV. A previous study by Takano et al. (2013) showed that hydrogen bonds are formed between the R118 residue of WT NA and OTV; however this was not observed in the previous study by Mhlongo & Soliman (2015) or in the current study. We speculate that this discrepancy is due to the trajectory used for analysis in the previous study by Takano et al. (2013), which was too short to adequately sample the conformational space of the system. Mhlongo & Soliman (2015) analyzed four distinctive 25-ns MD trajectories and suggested that the I117V mutation affects residue-residue interactions in NA that cause the drug-binding site to change its conformation, thereby altering residue-drug interactions between NA and OTV; however, the details were not clear. In the current study, we elucidated correlations between the residue-residue interaction of the R118-E119 pair and the residue-drug interaction of the D151-OTV pair in NA-OTV complexes by analyzing four distinctive 80-ns trajectories obtained from MD simulations, as shown in Fig. 4. As a result, we clarified the detailed molecular mechanism by which the I117V mutation in NA alters the inter-residue interactions between R118, E119, and D151, and destabilizes the residue-drug interaction between D151 and OTV, thereby reducing the susceptibility of NA to OTV.

We speculate that the I117V mutation not only affects the susceptibility of NA to OTV, but also viral fitness. With regard to viral fitness, we expect to extend the present study in the future to clarify the effects of the I117V mutation on the binding affinity of the natural sialic acid substrate to NA. However, according to Adams et al. (2019), viral fitness not only depends on the binding affinity between the substrate and the enzyme, but also on the catalytic efficiency of the enzyme. Hence, clarifying the catalytic reaction mechanism of NA using expensive computational methods, such as the QM/MM method

(*Sousa et al., 2017*), is required to study the effect of NA mutations on viral fitness. Since analyzing the catalytic reaction of NA is far beyond the scope of this study, we simply mention it here as a future subject.

### Designing a potential drug design against I117V mutant strains

As summarized in Table 1, the I117V mutation in NA reduces the binding free energy of NA to OTV by 2.72 kcal mol$^{-1}$, which corresponds to an approximate 100-fold decrease in the relative susceptibility of the I117V mutant NA to OTV compared to that of WT NA. The IC$_{50}$ value of OTV has been experimentally observed to change by a factor of 3–50 upon I117V mutation in NA (*Hurt et al., 2007*; *McKimm-Breschkin et al., 2007*; *Chen et al., 2010*; *Ilyushina et al., 2010*; *McKimm-Breschkin et al., 2013*; *Takano et al., 2013*; *Creanga et al., 2017*; *Kode et al., 2019*). Compared with substitutions that are selected under drug pressure and increase IC$_{50}$ by more than 600-fold, such as H274Y (*Hurt et al., 2012*), the I117V mutation does not dramatically affect susceptibility to OTV. Therefore, OTV treatment may be effective against the I117V mutant strain of the influenza virus. However, the I117V mutation may affect OTV resistance in synergism with other mutations. For example, *Hurt et al. (2012)* found that the introduction of the dual I117V + H274Y mutation in NA significantly decreased susceptibility to OTV (a 1,896-fold increase in IC$_{50}$) compared to that resulting from the H274Y mutation alone (a 650-fold increase in IC$_{50}$). Therefore, based on the new knowledge gained in this study, we propose guidelines for drug design that avoid the loss of drug sensitivity associated with the I117V mutation in preparation for the possible emergence of potent drug-resistant strains.

Based on our study, we suggest that an inhibitor with a longer positively charged group is better than one with a shorter positively charged group, such as the amino group in OTV, to avoid resistance from the I117V mutation that affects interactions between the inhibitor and the E119 and D151 NA binding site residues. A longer positively charged group in the inhibitor helps to reduce electrostatic repulsion between the negatively charged E119 and D151 side chains. For example, OTV has a short positively charged amino group that interacts with residues E119 and D151, while zanamivir has a long positively charged guanidino group. In fact, the I117V mutation in NA resulted in a significant 50-fold change in the IC$_{50}$ value for OTV but only a 1.6-fold change in the IC$_{50}$ value for zanamivir, which indicates that zanamivir is effective against the I117V mutant strain (*McKimm-Breschkin et al., 2013*). In this study, we used molecular simulations to understand the molecular mechanism of OTV resistance associated with the I117V mutation in NA in detail, which led to the establishment of new molecular design guidelines that effectively solve the drug resistance problem.

## CONCLUSIONS

In this study, we theoretically investigated the molecular mechanism of reduced OTV drug susceptibility in the A/H5N1 influenza virus harboring the NA I117V mutation using MD simulations.

In the WT NA-OTV complex, the interaction between R118 and E119 can play an important role in increasing the binding affinity of NA to OTV. In this case, the positively charged side chain of R118, located in the β-sheet region, can frequently interact with the negatively charged side chain of E119, preventing the electrostatic repulsion between E119 and D151. This enables both the negatively charged side chains of E119 and D151 to simultaneously form hydrogen bonding interactions with the positively charged amino group of OTV, thereby contributing significantly to the binding affinity between NA and OTV.

In the I117V mutant NA-OTV complex, the binding affinity of NA to OTV can be reduced by decreasing the opportunity for interaction between R118 and E119. In this case, the mutation reduces the tendency of R118 to form the β-sheet structure, leading to less frequent interaction between its positively charged side chain and the negatively charged side chain of E119. This increases the electrostatic repulsion between E119 and D151, making it difficult for both to simultaneously form hydrogen bonds with OTV, which in turn reduces the binding affinity between NA and OTV. Thus, after the I117V mutation in NA, influenza viruses are less susceptible to OTV because of changes in the residue interactions between R118, E119, and D151.

The present study has successfully clarified the molecular mechanism by which the I117V mutation in NA reduces the OTV drug susceptibility of the A/H5N1 influenza virus.

### Funding
This work has received financial support from the Joint Usage/Research Center Program at International Institute for Zoonosis Control, Hokkaido University, Sapporo, Japan. Mohini Yadav received support for living expenses and tuition fees from the Watanuki International Scholarship Foundation. The funders had no role in study design, data collection and analysis, decision to publish, or preparation of the manuscript.

### Grant Disclosures
The following grant information was disclosed by the authors:
Joint Usage/Research Center Program at International Institute for Zoonosis Control, Hokkaido University, Sapporo, Japan.
Watanuki International Scholarship Foundation.

### Competing Interests
The authors declare that they have no competing interests.

### Author Contributions
- Mohini Yadav performed the experiments, analyzed the data, performed the computation work, prepared figures and/or tables, authored or reviewed drafts of the paper, and approved the final draft.
- Manabu Igarashi conceived and designed the experiments, authored or reviewed drafts of the paper, and approved the final draft.

- Norifumi Yamamoto conceived and designed the experiments, analyzed the data, performed the computation work, prepared figures and/or tables, authored or reviewed drafts of the paper, and approved the final draft.

## Data Availability

The raw data is available in the Supplemental Files 1 and 2.

## Supplemental Information

Supplemental information for this article can be found online at http://dx.doi.org/10.7717/peerj-pchem.19#supplemental-information.

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
