# Peer review of "Theoretical insights into the molecular mechanism of I117V mutation in neuraminidase mediated reduction of oseltamivir drug susceptibility in A/H5N1 influenza virus"

_PeerJ Physical Chemistry, doi:10.7717/peerj-pchem.19_

## Round 0.1 · original submission · Minor Revisions

I see all comments by the reviewers as relevant and hope that you can revise the manuscript accordingly. I particularly encourage you to make sure to broaden the discussion as Reviewer 1 suggests including the suggestion by Reviewer 2 to think about a possible use of the described I117V mutation.
Thanks for submitting your manuscript.

Reviewer 1 ·

Basic reporting

This is a clearly written manuscript that highlights the relevant literature in the introduction. The figures are relevant, clearly labelled and of high quality. There was no raw data supplied, which would be the simulation trajectories and secondary data derived from trajectory analysis. These raw data would take a lot of space and re-analysis would be beyond the scope of this review.

Experimental design

The experimental design is evident of original primary research. It is made clear how the research question fills the knowledge gap of understanding how mutations outside the active site of an enzyme can influence the binding affinity of an inhibitor. The Methods are state-of-the art and described with sufficient detail that an expert would be able to replicate the methods.

Validity of the findings

All relevant data have been provided, however no statistical analysis was provided. It should be assessed, if the difference in hydrogen bond occupancy in figure 2 is statistically significant. At least error bars based on confidence intervals should be shown in figure 2 as well as in figure 3. The description of the hydrogen bond results in lines 213, 214 (and elsewhere) should distinguish between thermodynamic and kinetic effects by using the correct terms stable/unstable and inert/labile.
Relevant and clear conclusions are made, which are linked to the research question. However, the discussion lacks depth as pointed out in section 4.

Additional comments

The discussion is quite limited in scope. It is completely focussed at the results, without references to the wider literature and comparison with similar computational and/or experimental studies. Furthermore, the discussion should refer to the significance of the finding for the future prospects of OTV as an anti-influenza drug and any implications for future drug discovery. An appreciation that the mutation may affect viral fitness (with references to literature) as well as opportunities for further work should be shown. With regards to viral fitness, the present study may even be extended to include a comparison of the computed binding affinities of the natural sialic acid substrate to the mutant and wild-type enzyme.

·

Basic reporting

1. The manuscript is clearly written. Only in the introduction around lines 66-68, I would suggest to use the present tense, as the residues are still located at the described positions, this was not only the case when the structures were reported.

Experimental design

2. The authors report that they performed four independent simulations, with different starting conformations. How were these different starting conformations obtained?
3. I am typically not a fan of MM-PBSA calculations, but I laud the authors for the clear description of the methodology. Could they still say something about how the SA term was obtained and (more importantly) why they think a single-trajectory approach is appropriate? This is typically only the case if no conformational changes upon binding are to be expected. However, as the whole manuscript is about conformational changes due to the mutation, it is not unlikely that the indicated sidechains would perform very differently in an apo simulation of the protein? The argumentation that the triple trajectory approach leads to a lot of noise does not make the single-trajectory approach more appropriate.

Validity of the findings

4. At the end of the introduction, the authors make a rather bold statement that in previous studies the ‘simulations were too short to reach reliable statistical results’. This may be true, but after such a harsh statement, I would expect that the authors would go out of their way to show proper statistical tests for their analyses. As you have done four independent simulations, you could add error bars to the bar plots of figures 2 and 3. Is the decrease of the secondary structure elements of 12% or the change in the occurrence of hydrogen bonds by 10% or 30% actually statistically relevant?
5. For the free energy differences in table 1, the authors may want to emphasize what the difference in free energy is for an effect of a factor 3-50. This would be between 0.6 and 2.3 kcal/mol.
6. The authors may also want to emphasize that, based on table 1, they predict that the difference in binding affinity is mostly enthalpy driven and not entropy driven. Based on that analysis, it makes sense to focus on direct interactions, such as hydrogen bonds.

Additional comments

7. In what kind of interactions does D151 engage when it is not hydrogen bonded to OTV?
8. The authors may want to spend a few thoughts on how their newly derived knowledge on the mechanism of the I117V mutation can be put to practical use. Do they expect that drugs that rely less on the interactions with E119 and D151 would suffer less from this mutation? Could they give suggestions of such molecules?

---

## Round 0.2 · Minor Revisions

Please consider the reviewer's comments. The three remaining comments can be solved (I think) by textual changes in the manuscript. If you provide me with a pdf of the manuscript where your highlight the changes you made answering the reviewer's last comments, I will be quick in making a very likely positive decision.

·

Basic reporting

The authors have responded to my comments and have made modifications to the manuscript. It is basically acceptable for publications, but I would urge the authors to make a few small clarifications.

Let me follow my original comments:
1. OK

Experimental design

2. The authors write that the four independent simulations started from randomly selected conformations from the initial 10-ns equilibration. Were the velocities reassigned completely? If not, then you have actually performed the same simulation 4 times with a slightly different shift in starting time: picking a random configuration (positions and velocities) and starting a simulation is basically the same as continuing the simulation at that point. I guess you have reassigned velocities for the four simulations, but this is not stated in the manuscript.
3. The authors seem to brush away the argumentation that the single-trajectory approach in MMPBSA may not be appropriate when conformational changes are involved by stating that it is widely used and that it leads to results that match the experimental values. I would like to emphasize that this still does not make the single-trajectory approach appropriate. For the current manuscript and the aims of the authors, it may be sufficient, and I appreciate that the authors at least mention the possibility that the single-trajectory approach is sometimes not reliable. If they really want to make me happy, they would add ‘because it assumes that no significant conformational changes occur upon ligand binding’ to that statement.

Validity of the findings

4. OK
5. Here, I think that the authors should be more careful. The IC50 is a property that depends on the experimental setup (protein and substrate concentration), and the equation that the authors give (ΔG ~= RT ln IC50) is not correct. If experiments are done under the same conditions, however, a similar equation often does hold for the relative binding free energies: ΔΔG ~= RT ln [IC50(1) / IC50(2)]. This is also what the authors actually use in the manuscript.
6. OK

Additional comments

7. OK
8. OK

---

## Round 0.3 · accepted · Accept

Thank you for your patience!

·

Basic reporting

no comment

Experimental design

the authors have satisfactorily addressed my final concerns

Validity of the findings

the authors have satisfactorily addressed my final concerns

Additional comments

no comment